# Research and Engineering Application of Salt Erosion Resistance of Magnesium Oxychloride Cement Concrete

**DOI:** 10.3390/ma14247880

**Published:** 2021-12-20

**Authors:** Chenggong Chang, Lingyun An, Weixin Zheng, Jing Wen, Jinmei Dong, Fengyun Yan, Xueying Xiao

**Affiliations:** 1State Key Laboratory of Advanced Processing and Recycling of Non-Ferrous Metals, Lanzhou University of Technology, Lanzhou 730050, China; ccg168@isl.ac.cn; 2Key Laboratory of Comprehensive and Highly Efficient Utilization of Salt Lake Resources, Qinghai Institute of Salt Lake, Chinese Academy of Sciences, Xining 810008, China; zhengweixin@isl.ac.cn (W.Z.); wj580420@isl.ac.cn (J.W.); dongda839@isl.ac.cn (J.D.); 3Key Laboratory of Salt Lake Resources Chemistry of Qinghai Province, Xining 810008, China; 4College of Physics and Electronic Information Engineering, Qinghai University for Nationalities, Xining 810007, China; anlingyun0825@126.com

**Keywords:** magnesium oxychloride cement concrete, mechanical property, microscopic morphology, phase composition, salt erosion resistance

## Abstract

Aiming at the problem that ordinary cement concrete is subjected to damage in heavy saline soil areas in China, a new type of magnesium oxychloride cement concrete is prepared by using the gelling properties of magnesium oxychloride cement in this study, and the erosion resistance of the synthesized magnesium oxychloride cement concrete in concentrated brine of salt lakes is studied through the full immersion test. The effects of concentrated brine of salt lakes on the macroscopic, microscopic morphology, phase composition and mechanical properties of magnesium oxychloride cement concrete are investigated by means of macro-morphology, erosion depth, SEM, XRD and strength changes. The salt erosion resistance mechanism of magnesium oxychloride cement concrete is revealed. The results demonstrate that under the environment of full immersion in concentrated brine of salt lakes, there is no macroscopic phenomenon of concrete damage due to salt crystallization, and the main phase composition is basically unchanged. The microscopic morphology mostly changes from needle-rod-like to gel-like. Due to the formation of a new 5·1·8 phase on the surface layer and the increase in compactness, its compressive strength has a gradual increase trend. Based on the engineering application of magnesium oxychloride cement concrete, it is further confirmed that magnesium oxychloride cement concrete has excellent salt erosion resistance and good weather resistance, which provides theoretical support for future popularization and application.

## 1. Foreword

Saline soil areas are widely distributed in China, especially the Qinghai Chaerhan Salt Lake, which is a heavy saline soil area with a very high degree of salinization, and its main components are chloride and carnallite [1,2]. Ordinary cement concrete engineering buildings in this area under the heavy salt environment, the erosion is extremely serious, and after the completion of the construction, the surface of the concrete quickly shows powdered damage, which seriously affects the quality of the project and poses a serious threat to local production and life [3]. At present, the ordinary cement concrete treatment mainly consists of two methods: external coating and internal mixing. The external coating method is mainly to add epoxy resin-adding glass fiber reinforced plastic coating on the surface of concrete, or coating with asphalt and asphalt felt, or covering with anti-erosion paint and anti-seepage geotextile, to isolate the brine from immersion. The internal mixing method mainly adds chemical reagents or minerals in the concrete to reduce the water–cement ratio of the concrete and improve the compactness of the concrete, so as to resist the erosion of brine [4,5]. However, neither the external coating nor the internal mixing method fundamentally solve the salt erosion resistance of ordinary cement concrete in the saline soil area.

Qinghai province in China possesses extremely abundant magnesium chloride resources, with the existence of 4.82 billion tons of this mineral having been verified in this area. Most researchers apply magnesium chloride to produce a series of magnesium products including magnesium metal, anhydrous magnesium chloride, magnesium hydroxide, and magnesia whiskers, but these magnesium chlorides have not been fully exploited, and they accumulate for a long time, causing ‘magnesium damage’ and seriously affecting the surrounding environment. Recently, the development and utilization of magnesium oxychloride cement (MOC) has become an ecological, economic, and effective means of solving the problem of the accumulation of magnesium chloride [6].

Magnesium oxychloride cement, known as magnesium cement for short, was invented by the Swedish scholar Sorel in 1867 and was mainly composed of the MgO-MgCl_2_-H_2_O ternary system [7,8,9,10,11,12]. The magnesium oxychloride cement has lots of the advantages, for instance, early strength, good fire resistance, excellent abrasion resistance and low thermal conductivity [13,14,15,16], which is widely used in arts and crafts, composite boards and building materials and other fields [17,18,19,20], and is especially applied to the partition wall board having light weight, high strength, stable in performance, easy to transport, cut and installation, etc. When the new crown epidemic was raging around China, most simple temporary hospitals or isolated areas, such as Leishenshan Hospital and Huoshenshan Hospital (in Wuhan), were built with the partition wall board. Magnesium Oxychloride Cement Concrete (MOCC) uses the gelling properties of magnesium oxychloride cement to prepare a unique and new type of concrete by adding aggregates such as sand and crushed stone. It has the advantages of high strength in the early stage and in stable strength in the later stage [6].

At present, in the field of high-performance concrete and ordinary concrete, excellent scientific achievements have been achieved in durable properties, and erosion resistance to chloride salts [21,22,23,24]. However, there are few studies on salt erosion resistance of MOCC in the existing literature [25], especially in heavy saline soil areas. Therefore, in order to explore salt erosion resistance of MOCC, solve the erosion problem of engineering buildings in heavy saline soil areas, and make full use of the magnesium chloride resources in Qinghai Province, this paper prepares MOCC based on the gelling properties of magnesium oxychloride cement, and studies the influence of concentrated brine of salt lakes on the macroscopic and microscopic morphology, phase composition as well as mechanical properties of MOCC, exploring the erosion resistance and erosion mechanism of MOCC. At the same time, it is applied to engineering demonstrations to verify the erosion resistance of MOCC in a heavy saline soil area, enrich the theory about salt erosion resistance of MOCC, and provide a theoretical basis and demonstration reference for engineering applications.

## 2. Experiment

### 2.1. Raw Materials

(1). Bischofite, a white crystal, is produced in Golmud, China’s Qinghai province. Its main component is MgCl_2_·6H_2_O. The specific components are shown in Table 1.

(2). Table 2 lists the chemical composition of light burned magnesia, its main component is magnesia, chemical formula MgO, the activity is 50.51%, the place of production is Haicheng, China’s Liaoning province.

(3). Concentrated brine of salt lakes, potassium extraction by-product, extremely high concentration and complex composition, the main composition is MgCl_2_·6H_2_O, the specific composition is shown in Table 3.

(4). Sand, produced in China’s Qinghai province, belongs to medium sand, with a fineness modulus of 2.6; crushed stone, produced in Qinghai Province, with a particle size of 5–30 mm.

(5). Fly ash, produced in Datong, China’s Qinghai province, and its specific composition is shown in Table 4, the specific surface area is 430 m^2^/kg, and the apparent density is 2.4 g/cm^3^.

### 2.2. Preparation of MOCC

The MOCC was prepared according to the mixing ratio listed in Table 5, and the fall height was controlled at 30–50 mm, in which the molar ratio of active MgO and MgCl_2_ was 7.6:1. The specific operation process: (a) the 23.5 wt.% MgCl_2_ aqueous solution was prepared; (b) a certain amount of light burned magnesia, fly ash, sand, and crushed stone were added in turn to the mixer, accompanied by stirring evenly, and then 23.5 wt.% MgCl_2_ aqueous solution were added to them and stirred uniformly. After that, they were poured into the mold (150 mm × 150 mm × 150 mm) to harden, which were taken out after 24 h. Finally, they were cured naturally indoors. The indoor temperature was 22 ± 2 °C, and its humidity was 33 ± 5%. The experiments process was shown in Figure 1. The testing of the compressive strength was fixed as 1, 3, 6, 9 and 12 months, respectively.

### 2.3. Characterization

The MOCC that had been naturally cured indoors for 28 days were fully immersed in a water tank containing concentrated brine of salt lakes. At different ages, samples were taken in order to measure its strength, and the effect of concentrated brine of salt lakes on its erosion resistance was studied. Six pieces of the randomly selected samples were used for testing the compressive strength for each age by using electro-hydraulic pressure testing machine (model SYE-3000D, sanyuweiye testing machine Co., Ltd., Beijing, China). The compressive strength were obtained in accordance with Formula (1), and finally its average value as well as the standard deviation had been calculated.
(1)$$P=\frac{10\times F}{A}$$
where F was pressure, its unit was kN; A was area, the value of which was 225 cm^2^; P was compressive strength, whose unit was MPa.

The field emission scanning electron microscope (model SU8010, Hitachi High-Tech, Japan) was utilized to observe the microscopic morphology of MOCC before and after immersion in different ages. An X-ray diffractometer (model D8 Discover, BRUKER, Germany) was utilized to analyze the phase composition of the MOCC before and after immersion. The anode of the X-ray diffractometer was made of copper target, the scanning angle was 20–80°, and the scanning step length was 0.02°.

### 2.4. Engineering Application

MOCC was prepared according to the method described in Section 2.2, and the MOCC pavement was obtained based on “Code for Construction of Highway Pavement Base Course” (JTJ034-2000), “Technical Specification for Construction of Highway Cement Concrete Pavement” (JTG F30-2003) and “Cement Concrete Pavement Slipform Construction Code” (JTJ/T037.1-2000). At the same time, a tracking test was performed, and was characterized according to the characterization method described in Section 2.3 to verify the salt erosion resistance of MOCC.

## 3. Results and Discussion

### 3.1. Macro Morphology of MOCC

Figure 2 shows the macro photos of MOCC before and after immersion. It can be observed from Figure 2a that when the 12-month-old MOCC is naturally cured in indoor environment, its surface is not damaged due to dehydration and drying. At the same time, there is no dampness phenomenon and crystalline salt appearing on the surface of the concrete, proving that the molar ratio of active MgO and MgCl_2_ of concrete is appropriate. It can be seen from Figure 2b that no cracks appear on the surface of MOCC, which is fully immersed in the concentrated brine of salt lakes for 12 months. After fishing out the concrete from the concentrated brine of salt lake (Figure 2b), there are some white salt crystals on the surface of MOCC, as shown the arrow in Figure 2c, and they may be MgCl_2_·6H_2_O.

### 3.2. The Microscopic Morphology of the Surface Layer of MOCC

Figure 3 shows the microscopic morphology of MOCC before and after immersion in concentrated brine of salt lake at different ages. It can be seen from Figure 3 that the MOCC that are naturally cured for one month in the room show needle-like morphology, which are interconnected and staggered to form a crystalline network structure. At the age of 6 months, the needle-like morphology become thicker and larger, then at the age of 12 months, the needle-like morphology become thicker and adheres to each other, and some areas also appear gel-like morphology. This kind of gel state is relatively dense, which is conducive to the improvement of concrete strength. When the immersion time in concentrated brine of salt lake is 1 month, most of the microscopic morphology appears the short and thick rods. Shapes of needles and rods are flattened and interact with gel at 6 months, while for the age of 12 months, most of microscopic morphology are gelatinous.

### 3.3. Surface Phase Composition of MOCC

Figure 4 indicates the phase composition of MOCC before and after immersing in the concentrated brine of salt lake at different ages. It can be observed from Figure 4 that the MOCC are mainly composed of 5·1·8 (5Mg(OH)_2_·MgCl_2_·8H_2_O, P5) phases, MgO, Mg(OH)_2_, MgCO_3_, CaCO_3_, and SiO_2_ before and after immersion.

The formation process of the 5·1·8 phase is the active MgO in the MgCl_2_ aqueous solution first dissolves into Mg^2+^ and OH^−^ ions, and then they directly react with Cl^−^ and H_2_O in the slurry system to form 5Mg(OH)_2_·MgCl_2_·8H_2_O, as shown in Equation (2).
5MgO + MgCl_2_ + 13H_2_O → 5Mg(OH)_2_·MgCl_2_·8H_2_O(2)

The diffraction peak of MgO may be from the unreacted MgO, and Mg(OH)_2_ may be the hydration product of active MgO (as shown in Equation (3)), while CaCO_3_, SiO_2_ and MgCO_3_ are, respectively, derived from crushed stone, sand and uncalcined magnesite in MOCC. It can also be seen from Figure 4a,b that there are no obvious changes in the main phase components of MOCC, which are separately maintained under the natural indoor circumstance and immersed in concentrated brine of salt lake for 1, 6 and 12 months. This shows that the concentrated brine of salt lake has no observable effect on the phase composition of MOCC.
MgO + H_2_O → Mg(OH)_2_(3)

### 3.4. Erosion Depth of MOCC

Table 6 lists the erosion depth of MOCC immersed in concentrated brine of salt lake at different ages. It can be seen from Table 6 that at the age of 1 month, the erosion depth of MOCC is 5 mm, and the erosion depth remains basically unchanged during the later stage of the immersion, which is kept at 6 mm. There are no erosion marks appearing on the surface of the concrete sample. This shows that the extremely corrosive concentrated brine of salt lake solution has not completely immersed into the MOCC, and the MOCC shows excellent resistance to the erosion in the concentrated brine of salt lake solution.

### 3.5. Mechanical Properties of MOCC

Figure 5 shows the mechanical property of MOCC before and after immersion in concentrated brine of salt lake at different ages. It can be seen from Figure 5 that from 1 to 3 months the compressive strength of MOCC immersed in concentrated brine of salt lake are higher than the concrete which are naturally cured indoors. With prolonging the immerse time in concentrated brine of salt lake from 6 to 12 months, the compressive strength of MOCC is slightly lower than that of naturally cured in room. In addition, because of the growth of crystallization, the compressive strength of MOCC cured naturally during the 12-month period, increases gradually with the extension of curing time. Additionally, the compressive strength of MOCC at 12 month is 23.9 MPa higher than that of the 1-month-old age. After immersion, compressive strength of MOCC still tends to increase gradually, and the strength at the 12-month-old age is 13.3 MPa higher than that of 1-month-old age. This shows that the strength of MOCC does not decrease, but rises within 12 months when they are immersed fully in concentrated brine of salt lake, indicating strong salt erosion resistance.

The concentrated brine of salt lake solution penetrates into the MOCC after they are immersed in concentrated brine of salt lake for 1 and 3 months of age, and the erosion depth is 5–6 mm (see Table 6). At this depth, MgO that is not involved in the reaction and MgCl_2_ solution that come from concentrated brine of salt lake continue to react, forming a new 5·1·8 phase. The newly formed 5·1·8 phase can improve the strength of MOCC, and it is deposited on erosion layer, increasing the compactness of the layer. Therefore, the strength of MOCC immersed in concentrated brine of salt lake is higher than that of the concrete naturally cured in room at the same age. As the immersion time is further prolonged, the MgO on the surface of MOCC is completely reacted, and all the newly formed 5·1·8 phase is deposited on the surface of MOCC, and hence the erosion depth cannot be further increased. However, the crystallization of concrete in the internal layer still proceeds, but its growth is slightly slower. The strength of MOCC immersed in concentrated brine of salt lake is lower than the concrete, which is naturally cured in the indoor environment at the same age, but its strength is still improved, which is an increase of 13.3 MPa compared with 1-month-old age. At the same time, it can be observed from Figure 2 that there is a salt precipitation on the surface of MOCC after soaking, but there are no erosion cracks on it. This fully shows that the MOCC possess excellent salt erosion resistance.

### 3.6. Engineering Application Verification of MOCC

The Chaerhan Salt Lake in Qinghai province is a heavy saline soil area with a high surface salt content (50–80%). Most of them are magnesium chloride, carnallite and sodium chloride as well as a small amount of soil. In order to verify the salt erosion resistance of MOCC in saline soil area, the MOCC is prepared according to the aforementioned method, and MOCC pavement is built in the saline soil area of Chaerhan in Qinghai Province, as shown in Figure 6a. The macro morphology of MOCC (Figure 6b) and the test of tracking samples (Figure 6c) are analyzed, the salt erosion resistance of the MOCC pavement in the 12-month period is studied.

From the MOCC pavement shown in Figure 6a,b, there is no damage or protrusions being found on the road surface, and the overall structure is complete. The compressive strength of the tracking samples is listed in Table 7. It can be seen from Table 7 that the compressive strength of the MOCC tracking samples increases steadily with the extension of curing time. Compared with the concrete which is immersed in the concentrated brine of salt lakes, its compressive strength is lower, but their variation tendencies are the same.

SEM and XRD analysis are performed on the surface of MOCC tracking sample, as shown in Figure 7 and Figure 8. It can be seen from Figure 7 that, for the ages of 1 month and 6 months, the hydration products of MOCC tracking samples are mainly needle-shaped crystals, which are connected to each other to form a network structure. The needle-shaped crystals turn into flat shapes at 12 months, and most of them are in a gel state. It can be seen from Figure 8 that the diffraction peaks of the 5·1·8 phase and the Mg(OH)_2_ phase have no major changes in comparison with those immersed in concentrated brine of salt lakes. These suggest that the performance of the MOCC pavement is stable and the salt erosion resistance is better within 12 months. These also further confirm that the MOCC has excellent salt erosion resistance and good weather resistance.

## 4. Conclusions

To solve the problem that ordinary cement concrete is subjected to damage in the heavy saline soil areas, the erosion resistance of MOCC in concentrated brine of salt lakes is studied through the full immersion test. The effects of concentrated brine of salt lakes on the macroscopic, microscopic morphology, phase composition and mechanical properties of MOCC are examined. The main conclusions are as follows:(1)Under the extreme environment of full immersion in concentrated brine of salt lakes, the surface of MOCC is not damaged due to salt crystallization within a 12-month period, its phase composition does not change significantly, the main phase is still 5·1·8 phase, and its microscopic morphology is changed mainly from needle-like to gel-like. As the active MgO reacts with MgCl_2_ in concentrated brine of salt lakes again, forming a new 5·1·8 phase. It is deposited on the surface layer, making the surface layer more dense. As a result, the erosion depth does not deepen with the prolonged immersion time, and its compressive strength does not decrease but rises, showing good salt erosion resistance.(2)Based on the actual engineering application, the MOCC pavement is not destroyed, the mechanical properties are stable, the phase composition keeps unchanged nearly. The microscopic morphology changes to gel. These further verify that the MOCC has excellent resistance salt erosion performance and good weather resistance. This study provides theoretical and technical support for future promotion and application. Meanwhile this research can take full advantage of the large amount of magnesium chloride resources in Qinghai province, realizing the comprehensive utilization of magnesium resources and possessing excellent environmental effects.

## Figures and Tables

**Figure 1 materials-14-07880-f001:**
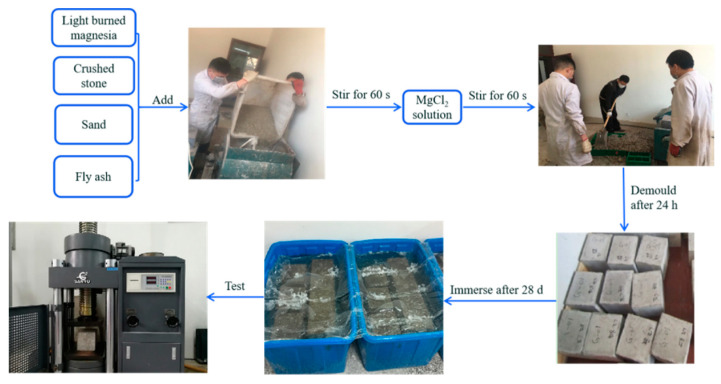
The experiments process of MOCC.

**Figure 2 materials-14-07880-f002:**
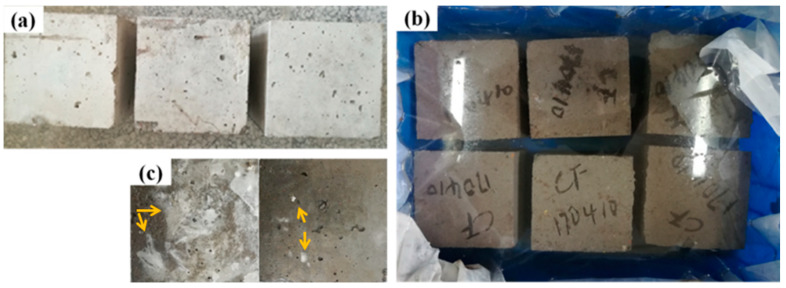
Macro morphology of MOCC (**a**) before and (**b**,**c**) after immersion in concentrated brine of salt lakes for 12 months. Among them, (**c**) is the case that MOCC presented in (**b**) is taken out from concentrated brine of salt lakes.

**Figure 3 materials-14-07880-f003:**
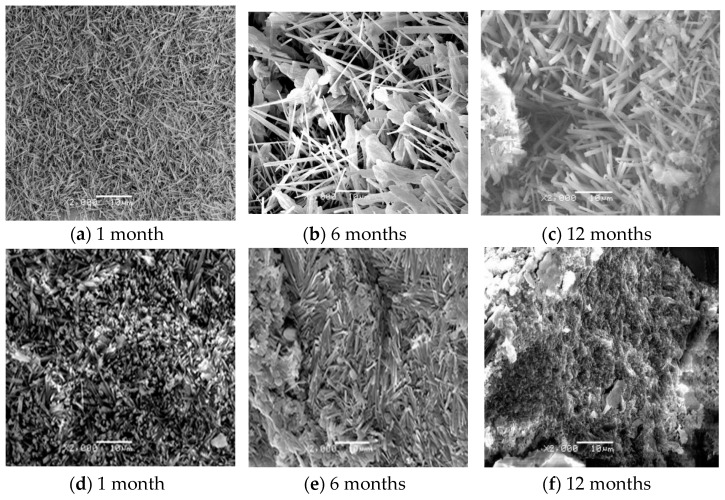
Microscopic morphology of MOCC (**a**–**c**) before and (**d**–**f**) after immersion in concentrated brine of salt lake for different months.

**Figure 4 materials-14-07880-f004:**
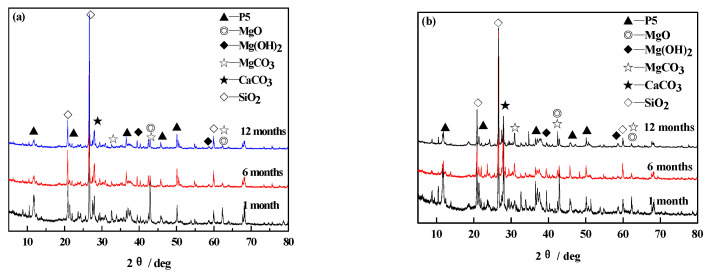
XRD patterns of MOCC (**a**) before and (**b**) after immersion in concentrated brine of salt lake for different months.

**Figure 5 materials-14-07880-f005:**
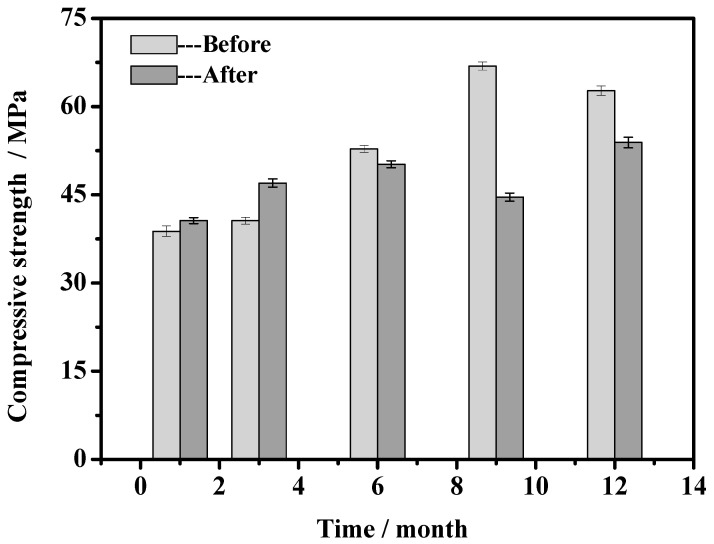
Mechanical properties of MOCC before and after immersion in concentrated brine of salt lake for different months.

**Figure 6 materials-14-07880-f006:**
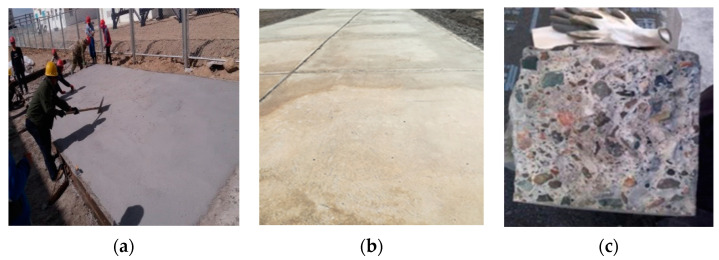
MOCC pavement and tracking sample (**a**) during construction process; (**b**) cured for 12 months; (**c**) cross section of tracking sample.

**Figure 7 materials-14-07880-f007:**
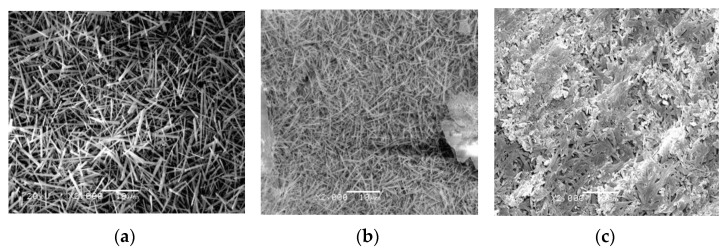
Microscopic morphology of tracking sample of MOCC for different months (**a**). 1 month (**b**). 6 months (**c**). 12 months.

**Figure 8 materials-14-07880-f008:**
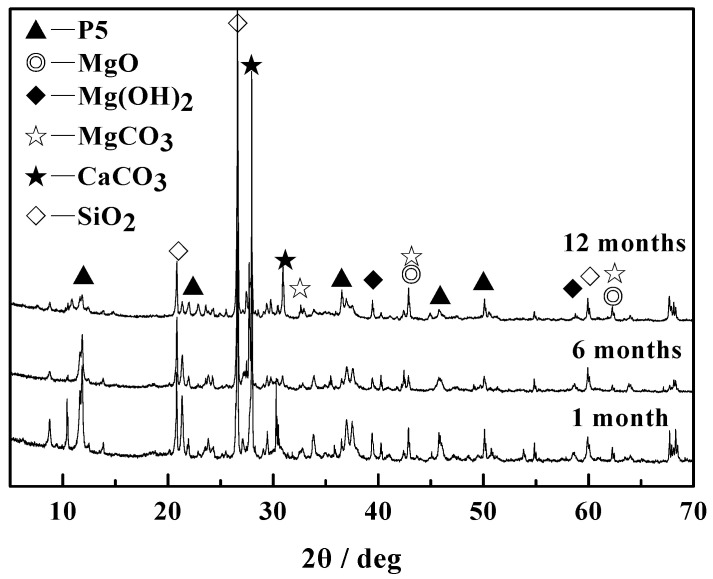
XRD patterns of tracking sample of MOCC for different months.

**Table 1 materials-14-07880-t001:** Chemical composition of bischofite.

Composition	MgCl_2_	NaCl	MgSO_4_	KCl	CaCl_2_	Water Insoluble Matter	H_2_O
Content (wt.%)	44.90	0.13	0.06	0.01	0.03	0.27	51.04

**Table 2 materials-14-07880-t002:** Chemical composition of light burned magnesia.

Composition	MgO	MgCO_3_	CaCO_3_	f-CaO	Acid Insoluble Matter
Content (wt.%)	69.52	19.80	1.34	0.38	8.41

**Table 3 materials-14-07880-t003:** Chemical composition/(wt.%) of concentrated brine of salt lakes.

Composition	K^+^	Ca^2+^	Mg^2+^	Na^+^	Fe^3+^	Al^3+^	Cl^−^	SO_4_^2−^	HCO_3_^−^
Content (g/L)	0.62	0.49	113.0	1.99	0.007	0.012	342.88	0.651	1.916

**Table 4 materials-14-07880-t004:** Chemical composition/(wt.%) of fly ash.

Composition	SiO_2_	Al_2_O_3_	K_2_O	CaO	Fe_2_O_3_	TiO	MgO
Content (wt.%)	40.04	41.87	3.29	1.49	9.68	0.95	0.74

**Table 5 materials-14-07880-t005:** Mixture ratio of MOCC.

Raw Materials	Light Burned Magnesia	Sand	Crushed Stone	Fly Ash	23.5% of Mass FractionSolution
Quality/Kg	12	40	63	1.8	7.9

**Table 6 materials-14-07880-t006:** Erosion depth of MOCC immersed in concentrated brine of salt lake for different months.

Soaking Time/Month	1	3	6	9	12
Erosion depth/mm	5	6	6	6	6

**Table 7 materials-14-07880-t007:** Compressive strength of tracking sample of MOCC.

Time/Month	1	3	6	9	12
Compressive strength/MPa	33.8	35.6	37.2	37.8	38.4

## Data Availability

The data presented in this study are available on request from the corresponding author.

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
