# Peer review of "Research and Engineering Application of Salt Erosion Resistance of Magnesium Oxychloride Cement Concrete"

_materials, 2021, doi:10.3390/ma14247880_

Round 1
Reviewer 1 Report
Comments on:
Chenggong Chang et al. Research and Engineering Application of Salt Corrosion Resistance of Magnesium Oxychloride Cement Concrete
- Line 13: usually the texts of papers are not personolized (my country).
- General: usually blanks are plced after points, commas, sem,icolons,…
- Table 1: rest is H2O??
- Line 89: ly ash: Please report particle size distributions and specific surface area.
- Line 93 and elsewhere: Be’ is not a SI unit. Density in g/cm³ should be used.
- Line 100: Please specify curing conditions (humidity, temperature).
- Line 104: Does that mean that for each parameter variation 6 samples were tested?
- Line 107: the parameter specification of equation (1) are barely understandable.
- Equipment used: please specify manufacturer and location of manufacturing.
- Line 132:is there any proof that the crysals are MgCl2.6H2O?
- Line 146: The reviewer cannot identify gel-like materials in the picture.
- Figure 2: Do the pictures show the surface of the sample? Or ist it a fracture surface?
- Line 156: what is phase 5.1.8: according to my view, this should be 5Mg(OH)2.MgCl2.8H2O? But please explain and report the chemical formula?
- Line 157: MgO
- Figure 3: It is advisable to apply Rietveld analysis of XRD patterns to yoield quantitative results.
- Table 6: How is the corrosion depth measured?
- Line 183: mechanical testing: Were the samples dried before testing? How?
- Figure 4: Please give some information on accuracy.
- The style of the references is quite unusual. Please check.
Reviewer 2 Report
The research topic is interesting. Please find some comments for further consideration:
Line 36: “Ordinary cement concrete engineering buildings in this area under the heavy salt environment, the corrosion is extremely serious, and after the completion of the construction, the surface of the concrete quickly appears powdered damage, which seriously affects the quality of the project and poses a serious threat to local production and life(Zhou et al.2011).”
The authors discuss the necessity of using MOC because of low corrosion resistance of ordinary cement concrete. However, the chloride in MgCl2 (main MOC ingredient) can cause a serious corrosion to the MOC structure itself by corroding the reinforcing steel bars. Therefore, I do not see any advantage of using MOC for the structures in brine environment of the salt lakes with high chloride concentration (as sown in Table 3).
Line 221: “MOCC pavement and tracking sample (a)construction; (b)12-month-old road surface; (c) Track the sample cross section”
Why the pavement color in 5a is different than 5b? It seems the photos are for different places and mixtures? What is the mix design of each pavement?
Line 225: “Table 6 lists the compressive strength of the tracking samples, can be seen from Table 6 that the compressive strength of the MOCC tracking samples increases steadily with the extension of time. Compared with the concrete that immerse concentrated brine of salt lakes, the compressive strength is less, but the change law is the same.”
These sentences are unclear. Please rephrase the sentences. If the leaching occurred on the MOC surface, it is unclear why the strength was increased!
Reviewer 3 Report
Dear authors ,
Thank you very much for the preparation of the article dealing with the modification of concrete in the area of salt-induced degradation.
The title and abstract are understandable and clear.
The introduction is short and insufficient. You need to expand it more with meaning, citations and other information and possibilities from the field of civil engineering.
I can recommend for example:
10.1016/j.conbuildmat.2020.119535
10.1016/j.conbuildmat.2021.123703
10.3390/ma14164706
The description of the materials and their properties is not bad and is probably sufficient.
What is missing is a broader testing methodology. The experiments are not sufficiently described.
The timing of testing is lacking - as with concrete it is important when the different tests are carried out.
Conclusions are too short.
The paper does not include all the mandatory items from the template.
References are not correct.
The article needs to be thoroughly corrected.
Round 2
Reviewer 1 Report
Line 129: kN
Table 6: Still no description found, how corrosion depth was determined
Reviewer 2 Report
Dear Authors,
Thanks for the reply. Unfortunately, I am still not convinced with your answer to the first question.
As the authors said, the MOCC was not deteriorated (corroded) more than 5-6 mm. Therefore, the decomposition of MOCC (authors called this process corrosion, which is not a precise term) is not determining factor for the deterioration of structures or pavements. However, the authors neglected the influence of chloride in MgCl2 solution on the corrosion of reinforcement which takes some years to be evident on the surface of pavement (even with using coatings on the steel surface).
The source of chloride in MOCC is internal. In other words, there is a huge amount of chloride inside the MOCC, sourcing from MgCl2 solution (the internal chloride content in MOCC can reach 7% by weight of concrete). Therefore, there is still a high amount of chloride inside the MOCC, even without chloride penetration from the lake into the MOCC. The presence of such a huge amount of chloride significantly corrodes the reinforcing steel and the structure or pavement will deteriorate after some years. The presence of a coating on the steel surface can only mitigate the corrosion rate of steel. For more information, please read textbooks/papers on the chloride threshold value for corrosion of steel in cementitious materials.
Reviewer 3 Report
Dear Authors,
the article is much better, but you have studied little literature on such an important subject.
Try to enrich your references with, for example, the following articles, expeciali in field of sustainibility:
10.3390/app11114964
10.3390/ma14164706
10.3390/ma14123316
You have improved the other things and I agree.
Regards,
